# Comparison of Chemical Composition, Physicochemical Parameters, and Antioxidant and Antibacterial Activity of the Essential Oil of Cultivated and Wild Mexican Oregano *Poliomintha longiflora* Gray

**DOI:** 10.3390/plants11141785

**Published:** 2022-07-06

**Authors:** Alma E. Mora-Zúñiga, Mayra Z. Treviño-Garza, Carlos A. Amaya Guerra, Sergio A. Galindo Rodríguez, Sandra Castillo, Enriqueta Martínez-Rojas, José Rodríguez-Rodríguez, Juan G. Báez-González

**Affiliations:** 1Facultad de Ciencias Biológicas (FCB), Universidad Autónoma de Nuevo León (UANL), Ave. Pedro de Alba S/N, Cd. Universitaria, San Nicolas de los Garza 66455, Mexico; alma_mora84@hotmail.com (A.E.M.-Z.); mayra_trevinogarza@hotmail.com (M.Z.T.-G.); carlos.amayagr@uanl.edu.mx (C.A.A.G.); sagrod@yahoo.com.mx (S.A.G.R.); sandra.castilloh@uanl.mx (S.C.); 2Department Analytics and Microbiologie, Zentrum fuer Ernährung und Lebensmitteltechnologie GmbH, 17033 Neubrandenburg, Germany; martinez@zelt-nb.de; 3Tecnológico de Monterrey, School of Engineering and Sciences, Monterrey 64849, Mexico

**Keywords:** oregano oil, antioxidant activity, Mexican oregano *Poliomintha longiflora*, wild oregano, cultivated oregano

## Abstract

Mexican oregano *Poliomintha longiflora* Gray located in the municipality of Higueras, Nuevo Leon, Mexico was collected during the autumn (September, OCO), winter (January, OCI) and summer (June, OCV) seasons, under cultivation conditions. It was also collected in wild conditions during the autumn (OSO). Essential oil (EO) was extracted from leaves and the color, refractive index and density were reported. The EO yield, antioxidant activity by ORAC assay, thymol and carvacrol concentration and antibacterial activity were statistically compared (*p*-value = 0.05). Among the various harvests, the highest EO yield, antioxidant activity, thymol and carvacrol content and antibacterial activity against *Salmonella* Typhi were observed in leaves harvested in autumn. In order to compare wild oregano with cultivated oregano, analyses were performed in the season with the highest essential oil yield and antioxidant activity, recorded in autumn. The main difference found was the ratio of thymol:carvacrol in wild oregano oil, which was 1:8.6, while in cultivated oregano, it was approximately 1:2, which was maintained in all three seasons. The EO on wild conditions showed the best antibacterial activity in *Salmonella* Typhi. On the other hand, wild and cultivated oregano showed similar antioxidant activity. One advantage of the use of cultivated oregano is that its supply is guaranteed, in contrast to that of wild oregano.

## 1. Introduction

Oregano is an aromatic plant which is widely used for culinary purposes and in traditional medicine for a variety of diseases [1], as well as for the extraction of its essential oils, which have recognized biological activity, including as an antioxidant and antibacterial additive in food [2,3,4]. The most important species are Greek oregano (*Origanum vulgare*), Spanish oregano (*Coridohymus capitatu*), Turkish oregano (*Origanum Onites*) and Mexican oregano (*Lippia berlandieri*) [3,5]. Among the Mexican oregano species is *Poliomintha longiflora* Gray, also called “smooth oregano”. This species is found in the northern part of Mexico, especially in the states of Coahuila, Tamaulipas and Nuevo Leon [5,6,7,8].

Most essential oils, including oregano oil, are classified as “generally recognized as safe” (GRAS) substances, which makes them potentially valuable for use in food, drugs and cosmetics [9]. The potential utility of the bioactive compounds present in oregano essential oil extends to the food industry, i.e., as an additive in functional foods, flavoring or nutraceuticals, as well as the agronomic industry, in perfumery and in pharmaceuticals. Recently, it has been studied as a potential inhibitor of COVID-19 [10,11].

It is known that the oil of *P. longiflora* Gray contains compounds with antibacterial and antioxidant properties. Several studies have shown the antioxidant potential of the essential oils of Greek oregano and common oregano, which is comparable to α-tocopherol and the synthetic antioxidant butylated hydroxytoluene (BHT), the latter of which has been progressively replaced by antioxidants from plants in food due to its adverse health effects [11,12,13,14]. The use of oregano essential oil as an antioxidant agent in food could represent a valid means to prepare food which is free of synthetic additives [2].

The antioxidant and antibacterial activities can be explained by the presence of compounds such as thymol and carvacrol. These phenolic compounds have hydroxyl groups [15], but it has been observed that when combined with other oxygenated monoterpenes, they can exhibit important antioxidant activities with synergistic behavior [2,16,17]. Another example of the use oregano essential oil is in broiler production, in which the application of antibiotic growth promoters has been prohibited in the European Union. In response, oregano essential oil has been implemented as a natural growth promoter and antibiotic [17,18].

On the other hand, the quality of oregano essential oil is dependent on its composition and concentration, and these variables depend in turn on the genotype, climatic and geographical factors, periods of light, temperature, water stress, altitude, harvest time, soil composition and growth stage [1,19,20,21]. The yield and composition of oregano essential oil vary based on these factors [22].

Worldwide, Mexico is the second largest producer of oregano, mainly with the *species L. graveolens* Kunth, *L. berlandieri* Schauer, and recently, *P. longiflora* Gray [8,10,23], but due to the fact that most of the commercially exploited species are of wild origin and depend on the rainy season for their harvest, as well as being also affected by increasing climatic changes such as long dry seasons [5,23], neither the supply of oregano nor the quality of its essential oil is ensured. Moreover, the natural recovery of this resource is not guaranteed. As such, one strategy to avoid the decrease of wild oregano populations is the use of commercial crops. The production of cultivated oregano would generate changes in the composition of the essential oil with respect to wild oregano but would stabilize supply. Therefore, it is necessary to evaluate the yield and quality of the essential oil of cultivated oregano [24] if it is to be used meet the increasing demand for bioactive compounds from its essential oil [10].

The main aims of this work were to evaluate the quality in terms of yield, thymol and carvacrol content as well as the antioxidant activity of the essential oil of a cultivated oregano *Poliomintha longiflora* Gray during the winter, summer, and autumn seasons, and to compare the EOs of cultivated and wild oregano in autumn, since this is the rainy season and therefore, the period when the highest yields.

## 2. Results and Discussion

### 2.1. Physicochemical Properties

The physicochemical properties of color, refractive index and density of the essential oils of cultivated and wild oregano are shown in Table 1 and compared with anethole oil as a standard reference. The density and the refractive index (R.I.) were similar for all oils. The RI and density reported Maree and Ian (2000) for *B. anisata* oil (89.9% anethole) were 1.5572 and 0.984, respectively, similar to the values obtained for anethole [25].

The results obtained for the oil yield of oregano show that significant differences (*p* < 0.05) occurred according to season, with higher oil yield obtained in autumn season, i.e., 2.8%, compared to winter and summer, i.e., 1.7% and 1.9%, respectively. The autumn season is characterized by the presence of rains, and the increase in yield may be due to this factor. The literature reports an increase in the production of essential oil in the rainy season [26]. On the other hand, this increase also coincides with that reported for the oregano species *Poliomintha* by Rodriguez et al. [8], in which the highest yield of essential oil of oregano grown in the municipality of Higueras was obtained during the autumn. These facts explain why EO in wild conditions is more abundant than in cultivated oregano in autumn, as is shown in the Table 1.

In contrast, according to some authors, the differences in the yields of essential oils are mainly due to the origin [27], with values from 1% to 5.6% having been reported for oregano species [28,29,30].

### 2.2. Content of Thymol and Carvacrol and Chemical Composition of Essential Oils

The thymol content (%*w*/*w*) in oregano oils was in the range of 14.4–16.2%, with no significant difference (*p*-value = 0.05) found in the three seasons (Table 2). In wild oregano, the thymol content showed a significantly lower value, with 5.9%; on the other hand, in cultivated oregano oil, the carvacrol content was the highest in the autumn; the same was true for wild oregano oil, with 50.9%. Therefore, autumn is the most logical time to harvest the plant due to the highest oil yield and the highest concentrations of thymol and carvacrol.

In studies on samples from the aerial parts (leaves and inflorescences) of wild *P. longiflora* Mexican oregano from Chihuahua, 33.2% thymol and 17.6% carvacrol were reported [3]. These results are different from those found in cultivated oregano in our work (thymol 16.2% and carvacrol 32.6% in the autumn), but for *O. dictamnus* oregano cultivated in Chihuahua, a carvacrol content of over 47% and the absence of thymol were reported [19]. In the sum of the terpenoid thymol and carvacrol (%*w*/*w*), a significant difference was found in the autumn, which showed the highest total content of these compounds, i.e., 48.8%, as shown in Table 2. On the other hand, the ratio of thymol and carvacrol in cultivated oregano oil was in the range of 1:1.6 to 1:2.1, that is, for 1 part thymol, there were approximately two parts carvacrol, as shown in the Table 2.

Wild oregano had the highest ratio of thymol:carvacrol, i.e., 1:8.6, compared to cultivated oregano. This means that wild oregano oil has much more carvacrol with respect to thymol than cultivated oregano. Regarding the main phenol contents, i.e., thymol and carvacrol, wild oregano presented 56.9%, which is significantly higher than those obtained under cultivation conditions. As suggested by Walczak et al., a high carvacrol content is desirable when looking for specific biological activity [31]. In their study, the influence of thymol and carvacrol on biofilm surfaces of polyvinyl chloride, polypropylene, polyethylene, and stainless steel were studied. It was found that carvacrol was more efficient than thymol at reducing biofilms. On the other hand, several studies reported that the antioxidants present in oregano are found in different amounts and distributions, depending on the part of the plant, such as leaves, stems, and flowers [30]. In the leaves of *Lippia origanoides*, the following quantities were reported: carvacrol + thymol, 53.2%; ρ-cymene: 10.1%; ϒ-terpinene and trans-β-caryophyllene: 3.4% [28]. Another study reported that in six essential oils of oregano (*Origanum vulgare*) analyzed by GC/MS, thymol and carvacrol [32] were detected in quantities of 0.2–5.8% and 58.7–77.4%, respectively.

In our study, the composition of the essential oil from cultivated oregano oil harvested in summer and autumn was analyzed by GC-MS. Four chemical compounds were predominant, which, in decreasing order, were carvacrol, thymol, gamma terpinene and carvacrol methyl ether. The sum of the components represented 78.07% and 75.61%, respectively, of the oil composition, as can be seen in Table 3. In the winter season, the sum of carvacrol, thymol, carvacrol methyl ether and gamma terpinene represented 70.55%.

In wild oregano, the predominant chemical components in decreasing order were carvacrol, gamma terpinene, thymol and carvacrol methyl ether, with the sum of these components representing 78.76%. The main difference between cultivated oregano and wild oregano is the high content of carvacrol in the latter, representing more than half of its total composition, while in cultivated oregano, this compound represents one third of the total composition. In a study conducted by Sarrazin et al. [33], the composition in oregano Lippia origanoides was reported in the rainy and dry seasons; carvacrol was the main component, comprising 43.5% and 41.4%, respectively, followed by thymol, with 10.7% and 10.6%, ρ-cymene with 9.8% and 10.0%, and ρ-methoxythymol with 9.6% and 10.4% respectively, for a total of 78.6% of the oil composition. Little difference was observed among the seasons.

### 2.3. Antioxidant Activity

Since the antioxidant potential of oregano oil is widely recognized, only one method was used to compare the various oils in this study, namely, the ORAC assay, because it employs a biological radical (peroxyl radical) in its reaction. The results obtained were expressed as ORAC units and as the % Trolox equivalent (%TE), where a Trolox value (400,000 µmol/100 g) was assigned a reference value of 100%.

The results obtained as ORAC units (micromoles of Trolox equivalent (TE) antioxidant activity per 100 g) showed values ranging from 121,350 to approximately 153,380, as shown in Table 4. No significant differences were found in the ORAC values observed in cultivated oregano oil during the summer and autumn. The lowest ORAC (121,350 µmol TE/100 g EO) value was observed during the winter.

The results obtained in the comparison by origin of wild and cultivated oregano showed no significant difference, with ORAC values of 153,380 and 142,460, respectively. When the results were expressed as % TE, the cultivated oregano oil reached 38.3% and wild oregano oil 35.6% of the antioxidant capacity of Trolox, as shown in Table 4. These results indicate that during the ORAC assay, all cultivated and wild oregano oils were able to neutralize free radicals to at least 30% with respect to the performance of Trolox. Other studies [17,34] on Greek oregano oil and in common oregano oil reported the equivalent of 8.5% and 8.5% of Trolox performance, with 34,008 and 34,296 µmol/100 g. Another study from 2020 [35] reported an ORAC value of 4147 for Greek oregano and 5933 for Spanish oregano [36]. In addition, in a survey by Zhen W. et al. [37], oregano *Poliomintha* showed the highest ORAC value among 39 analyzed herbs.

Another study [36] on oregano essential oil (*Origanum vulgare* L.) used in encapsulation processes reported ORAC values of 208,000 ± 2100 µmol TEAC/100 g, equivalent to 52% of the performance of Trolox. On the other hand, the USDA database [38] reports an ORAC value of 159,277 for oregano. According to the results shown by Carrasco et al., for essential oils from cultivated oregano, the ORAC value ranges were variable [32]. In their study, six different essential oils of cultivated *Origanum vulgare* showed the equivalent of 140,000 to 250,000 ORAC values. Also, in extracts of 42 varieties of cultivated and wild oregano [39], the lowest ORAC value, i.e., 159,000, and the highest, i.e., 339,000, were reported. The results obtained in our study suggest that the antioxidant activity identified in the oil of oregano *Poliomintha longiflora* remained without significant changes in the summer and autumn seasons. This means that the EO may be harvested at those times without risk of significant changes in its biological antioxidant activity.

### 2.4. Antibacterial Activity

The antibacterial activities of oregano oils were tested against four foodborne pathogens. *S. aureus*, *S.* Typhi, *L. monocytogenes* and *B. cereus* were selected due to their prevalence in food products and the fact that they are representative of the two main groups of pathogens, i.e., Gram (+) and Gram (−). The assay was conducted according to the National Committee For Clinical Laboratory Standards (NCCLS) in Muller Hinton agar. The antibacterial activity was evidenced in all oils evaluated with inhibition zone diameters ranging from 3.1 to >5.4 cm (Table 5). In cases where the halos overlapped, >5.4 was reported, since it was the largest diameter that could be measured. The inhibition zones obtained from oregano oils were compared with gentamycin as a positive control and dicloxacillin as a negative. The strains used in this study showed susceptibility to gentamycin, with haloes ranging from 2.1 to 1.7 cm, with no significant differences among the strains. On the other hand, the strains showed resistance to dicloxacillin, with no inhibition zones surrounding the disks or with a minimum diameter of 0.9 cm for *L. monocytogenes,* which according to NCCLS standards, is sufficient for it to be considered a resistant strain. The antibacterial activity shown by the oils was higher than that of the gentamycin. Significant differences in susceptibility were evidenced among strains, with *B. cereus* and *L. monocytogenes* being more susceptible than *S.* Typhi and *S. aureus*, with diameters of inhibition >5.4 cm for all oregano oils tested. On the other hand, significant differences in antibacterial activity were observed among the types of oregano oils for the other two strains tested, i.e., OSO and OCO, being the oils with the greatest antibacterial activity against *S.* Typhi and OCI for *S. aureus*. Previous studies have reported more resistance in bacteria Gram (−) than Gram (+) due to the differences in their cell walls. Additionally, the antibacterial response between bacteria of the same Gram could vary due to the type and number of components present in each oil and a possible synergy among them [40]. Our results suggest that these oils inhibited the growth of the test microorganisms at low concentrations, and as such, could be used in the food industry, although more specific studies are suggested.

## 3. Materials and Methods

### 3.1. Plant Material

Oregano *Poliomintha longiflora* Gray was grown in the Municipality of Higueras, Nuevo León, Mexico, located at latitude: 25°58′20.88″ N, longitude: 100°0′54.00″, with an altitude of 490 m above sea level and annual rainfall of 200 to 450 mm, according to INIFAP. The initial plants were collected in their wild habitat in the same locality and open sky cultivation conditions, with irrigation canal installations, using chicken manure as fertilizer. The aerial parts of the plants were collected in a single day in the winter (January), summer (June) and autumn (September). Additionally, the wild type was sampled on the same day as the cultivated samples from the autumn season. In all cases, sampling was carried out in the same municipality. Plant specimens were identified by the Ecology Department of the Facultad de Ciencias Biológicas, UANL.

### 3.2. Essential Oil

The essential oil was obtained by a hydro distillation process [41] using a modified Clevenger type apparatus for 45 min from dry samples of 300 g of leaves. Extractions were performed in triplicate. The obtained oils were isolated using a separatory funnel, dried with anhydrous sodium sulfate and stored in amber vials in freezing conditions until further analysis [42]. The oil was weighed to determine its yield [15].

### 3.3. Density, Color and Refractive Index of Oregano Oil

Oil density was determined using a mass over volume pycnometer. The color was determined visually, and for the determination of the refractive index, an Abbemat 300 refractometer (Anton paar, Graz, Austria) was used according to a method described in the literature and ISO 279:1998, ISO 280:1998 [15,43].

### 3.4. Determination of the Content of Thymol and Carvacrol and Chemical Composition by Gas Chromatography-Mass Spectrometry (GC-MS) Analysis

The oregano essential oil samples were diluted in hexane; the injected volume was 1 µL. To quantify the thymol and carvacrol contents, external standards were used. A calibration curve was made in the range of 6.2–49.6 mg/L for thymol, with a correlation coefficient of r^2^ = 0.9965, and 10.9–87.2 mg/L for carvacrol, with a correlation coefficient of r^2^ = 0.9977. An Agilent 5975C gas chromatograph (Agilent Technologies, Santa Clara, CA, USA) was used under the following conditions: capillary column HP-5 MS (5% phenyl/95% dimethylpolysiloxane) (30 × 0.25 mm × 0.25 μm) from J & W Scientific (Folsom, CA, USA). The temperature ramp was as follows: Split injector (1:15) mode, helium as gas carrier at 1 mL min^−1^, oven at 45 °C for 1 min, increasing to 240 °C at 5 °C min^−1^ and holding for 4 min, interface temperature 250 °C, electron ionization at 70 eV, quadrupole mass analyzer with a mass range of 20–350 m/z. The identification of the compounds was carried out by comparing their mass spectra with those of the Wiley 7 n.L library, considering a quality coincidence >85%.

### 3.5. Antioxidant Activity by Oxygen Radical Absorbance Capacity (ORAC) Assay

The determination of antioxidant activity was carried out based on the methodology described by Prior [44] with slight modifications. Calibration curves were performed with 6-Hydroxy-2,5,7,8-tetramethylchroman-2-carboxylic acid (Trolox) in a concentration range of 6.25–100 μM in saline solution. Antioxidant capacity results were expressed in micromoles Trolox equivalents per 100 g of essential oil (μM TE/100 g) [45]. The determinations were performed in a 96-well microplate. First, 25 μL of Trolox standard or oregano oil sample diluted in methanol (50 mg/L) was placed in each well. Then, 150 μL of 0.02 µM fluorescein was added [46] and the mixture was incubated in the dark at 37 °C for 20 min. After this time, 25 μL of 2,2-Azobis-2-methylpropionamidine dihydrochloride (AAPH) 153 mM was added and the mixture was shaken. Then, fluorescence was measured using a BioTek Synergy 2 model microplate reader (BioTek Instruments, Winooski, VT, USA). The areas under the curve of the fluorescence intensity versus time plots were calculated using KC4 TM Data reduction Software (BioTek instruments, Winooski, VT, USA).

### 3.6. Antibacterial Activity

Four bacterial strains were selected to evaluate the antibacterial activity of oregano essential oils: *Salmonella* Typhi ATCC 19430, *Bacillus cereus* ATCC 13061, *Staphylococcus aureus* ATCC 25923, *Listeria monocytogenes* ATCC 7644. These were obtained from an American type culture collection. The strains were kept at −80 °C in brain heart infusion broth (BHI) with 20% *v/v* glycerol (Difco Laboratories, Sparks, MD, USA). An aliquot (50 µL) was taken from the original frozen culture and added in tubes containing 5 mL of the respective broth. Fresh cultures (18 h) were obtained in Mueller Hinton broth (MH, Difco Laboratories) or trypticase soy broth for *L. monocytogenes* (TS, Difco Laboratories) for further analyses. The antibacterial activity was determined by the disk diffusion method, as described by Rostro-Alanís et al. [40], and the NCCLS standard. Aliquots (100 μL) of fresh and adjusted cultures (0.1 OD 600 nm) (~1 × 10^6^) of each bacterium were homogeneously scattered in petri dishes containing Muller Hinton or trypticase soy agar. After that, filter paper disks (Whatman No. 1) of 6 mm in diameter impregnated with 10 µL of each oregano oil type were placed on the agar surface. The plates were incubated at 37 °C for 24 h or 30 °C for *L. monocytogenes*. The antibacterial activity was evidenced by the absence of microbial growth. The diameter of inhibition zones surrounding the disks was measured with a vernier caliper. Disks of gentamicin (10 µg) were used as a positive control, while dicloxacillin was employed as a negative.

### 3.7. Statistical Analysis

The analysis of results was performed by ANOVA using the statistical package SPSS (IBM, Armonk, NY, USA) (21.0), and Tukey’s test, considering significant differences at *p* < 0.05.

## 4. Conclusions

The applications of essential oils are numerous, and new technologies will further broaden and diversify their use, such as encapsulation through emulsions. Additionally, combining them with other substances will give rise to synergisms. The results obtained in cultivated oregano suggest that during the summer and autumn seasons, the antioxidant activity remains unchanged, while the thymol and carvacrol ratios are maintained during summer, autumn and winter. The results also showed that the autumn season favored the synthesis of the bioactive compounds carvacrol and thymol. Also at this time, cultivated and wild oregano were similar in terms of their antioxidant activity and in the sum of carvacrol + thymol. The main difference found was the ratio of thymol:carvacrol in wild oregano oil, i.e., 1:8.6, while in cultivated oregano, it was approximately 1:2; this was maintained in all three seasons. The major components of the essential oils were carvacrol, thymol, gamma terpinene and carvacrol methyl ether. Unlike wild oregano, one of the advantages of using cultivated oregano is that it makes it possible to achieve semi-standardized production in terms of the thymol and carvacrol contents. Another advantage is that production of cultivated oregano is a sustainable process which contributes to the reduction of deforestation and the maintenance of wild oregano populations. A large difference between cultivated and wild oregano is the high content of carvacrol in the latter, i.e., above 50%. Despite this difference, the biological activity did not show marked differences, that is, the antioxidant activity was similar in both oreganos, and although, in some regards, the antibacterial activity of wild oregano was better, there were more similarities than differences between cultivated and wild oregano. As such, the production of cultivated oregano is recommended to obtain essential oil.

## Figures and Tables

**Table 1 plants-11-01785-t001:** Physicochemical properties: color, refractive index, density and essential oil yield in cultivated and wild oregano oil.

		Physicochemical Properties		
EO	Color	R.I.	Density (g/mL)	Essential oil yield (%*w*/*w*)
OCI	Light yellow	1.4618 ± 0.0000 ^a^	0.9277 ± 0.0012 ^a^	1.76 ± 0.2 ^b^
OCV	Light yellow	1.4619 ± 0.0000 ^a^	0.9207 ± 0.0012 ^a^	1.91 ± 1.2 ^b^
OCO	Light yellow	1.4616 ± 0.0000 ^a^	0.9233 ± 0.0012 ^a^	2.81 ± 0.2 ^a^
OSO	yellow	1.4618 ± 0.0000 ^a^	0.9287 ± 0.0006 ^a^	3.5 *
A	Light yellow	1.557 ± 0.0000 ^a^	0.983 ± 0.0010 ^a^	

OCI: winter cultivated oregano; OCV: summer cultivated oregano; OCO: autumn cultivated oregano; OSO: autumn wild oregano; A: Anethole. * Single extraction, lowercase letters represent no significant differences (*p* < 0.05) between EOs.

**Table 2 plants-11-01785-t002:** Content of Thymol, Carvacrol, Sum of Thymol and Carvacrol (%*w*/*w*) and Ratio Thymol:Carvacrol in cultivated and wild oregano oil.

EO	Thymol	Carvacrol	Sum Thymol and Carvacrol	Ratio Thymol:Carvacrol
OCI	14.67 ± 0.8 ^a^	23.60 ± 1.9 ^c^	38.3 ± 2.7 ^b^	1:1.6 ± 0.1 ^b^
OCV	14.47 ± 0.9 ^a^	27.11 ± 1.4 ^cb^	41.6 ± 2.3 ^b^	1:1.9 ± 0.0 ^b^
OCO	16.20 ± 1.4 ^a^	32.63 ± 2.6 ^b^	48.8 ± 4.0 ^ab^	1:2.0 ± 0.0 ^b^
OSO	5.91 ± 0.8 ^b^	50.96 ± 8.1 ^a^	56.9 ± 8.7 ^a^	1:8.6.0 ± 0.8 ^a^

OCI: winter cultivated oregano; OCV: summer cultivated oregano; OCO: autumn cultivated oregano; and OSO: autumn wild oregano. Lowercase letters represent no significant differences (*p* < 0.05) between EOs.

**Table 3 plants-11-01785-t003:** Average composition in relative areas, % (*n* = 3) of volatile compounds in leaf samples of cultivated and wild oregano oil.

	Wild Oregano			
	Winter	Summer	Autumn	Autumn
**Monoterpene Hydrocarbons (MH)**				
Gamma terpinene (MHGT)	10.14	13.30	13.70	9.71
**Monoterpene Oxygenated (MO)**				
Borneol (MOBO)	3.36	3.53	2.82	1.65
Carvacrol methyl ether (MOCM)	14.85	8.82	5.05	4.27
Thymol (MOTI)	17.29	19.29	16.97	5.53
Carvacrol (MOCA)	28.27	36.66	39.89	59.25
Carvacryl acetate (MOCL)	1.82	1.59	3.12	1.84
**Sesquiterpene Hydrocarbons (SH)**				
*Trans*-Caryophyllene (SHTC)	ND	2.51	2.10	1.63
Total of majority components	75.73	83.19	83.65	83.88
Others componentes	24.27	16.81	16.35	16.12
**MH**	10.14	13.30	13.70	9.71
**MO**	65.59	69.89	67.85	72.54
**SeH**	0.00	2.51	2.10	9.71
**Sum** (MHGT + MOCM + MOTI + MOCA)	70.55	78.07	75.61	78.76

Compounds listed in order of chromatography elution.

**Table 4 plants-11-01785-t004:** Antioxidant activity in ORAC units and % TE in cultivated and wild oregano oil.

	Antioxidant Activity	
EO	ORAC unit *	% TE **
OCI	121,350 ± 4500 ^b^	30.3 ± 3.7 ^b^
OCV	142,520 ± 8600 ^a^	35.6 ± 6.0 ^a^
OCO	153,380 ± 2400 ^a^	38.3 ± 1.5 ^a^
OSO	142,460 ± 8000 ^a^	35.6 ± 5.6 ^a^

* 1 ORAC unit = 1 μmol of TROLOX/100 g. ** TE = Trolox equivalent percentage; this is the percentage of antioxidant activity reached by the EO equivalent to Trolox with respect to a reference value of 100% (400,000 μmol/100 g). OCI: winter cultivated oregano; OCV: summer cultivated oregano; OCO: autumn cultivated oregano; and OSO: autumn wild oregano. The results are expressed as mean ± standard deviation. Lowercase letters represent no significant differences (*p* < 0.05) between EOs.

**Table 5 plants-11-01785-t005:** Disk diffusion assay in cultivated and wild oregano oil.

	Inhibition Zone (cm)			
EO	*S.* Typhi	*B. cereus*	*S. aureus*	*L. monocytogenes*
OCI	3.3 ± 0.9 ^bc^	>5.4	4.7 ± 0.3 ^d^	>5.4
OCV	3.2 ± 0.2 ^b^	>5.4	4.0 ± 0.3 ^c^	>5.4
OCO	3.9 ± 0.1 ^c^	>5.4	3.1 ± 0.1 ^b^	>5.4
OSO	5.4 ± 0.2 ^d^	>5.4	3.5 ± 0.7 ^bc^	>5.4
Gentamycin	1.9 ± 0.1 ^aA^	2.1 ± 0.0 ^A^	1.7 ± 0.3 ^aA^	2.1 ± 0.1 ^A^
Dicloxacillin	NI	NI	NI	0.9 ± 0.1

OCI: winter cultivated oregano; OCV: summer cultivated oregano; OCO: autumn cultivated oregano; OSO: autumn wild oregano; NI: no inhibition. Lower and uppercase letters represent no significant differences (*p* < 0.05).

## Data Availability

Not applicable.

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
