# Peer review of "Comparison of Chemical Composition, Physicochemical Parameters, and Antioxidant and Antibacterial Activity of the Essential Oil of Cultivated and Wild Mexican Oregano Poliomintha longiflora Gray"

_plants, 2022, doi:10.3390/plants11141785_

Round 1
Reviewer 1 Report
In this manuscript, the authors reported that “Comparison of Chemical Composition, Physicochemical Parameters, Antioxidant, and Antimicrobial Activity of the Essential Oil of Cultivated and Wild Mexican Oregano Poliomintha longiflora Gray”. In my opinion, the manuscript investigates some important biological properties for use of essential oil from cultivated plants. However, it needs to be addressed in several comments.
Some critical suggestions
1. In the title, "antimicrobial” activity should be corrected into antibacterial. Also, “antimicrobial” is to be removed and replaced with “antibacterial” throughout the whole manuscript. Because this manuscript only tested the inhibitory activity against bacteria but not included other microorganisms.
2. Abstract: Line no. 29, “3 seasons” corrected into “three seasons”.
3. The table caption for figure 4 should be corrected. How author mentioned the sum….. of the ratio?
4. Table 5. Why wild oregano was not compared with all three seasons? Please explain and correct Winter O.
5. The zone of inhibition is very high, and the oil is loaded into the 6 mm disks. How was the zone of inhibition calculated? The authors suggested providing visual evidence for the antibacterial activity of the oil.
6. Overall, the comparison is useful for that condition. But those finding are not considered useful widely. If so, please explain in the introduction.
7. In conclusion, the author mentioned that “The main difference found was the ratio of thymol: carvacrol in wild oregano oil was 1:8.6 while in cultivated oregano, it was approximately 1:2 which was maintained in all three seasons”. How was it calculated?
Author Response
RESPONSE TO REVIEWER 1
Dear Reviewer #1.
We appreciate your valuable time and comments on the manuscript entitled “Comparison of chemical composition, physicochemical parameters, antioxidant, and antimicrobial activity of the essential oil of cultivated and wild Mexican oregano Poliomintha longiflora Gray”. The answers were enumerated as a below appointed.
- In the title, "antimicrobial” activity should be corrected into antibacterial. Also, “antimicrobial” is to be removed and replaced with “antibacterial” throughout the whole manuscript. Because this manuscript only tested the inhibitory activity against bacteria but not included other microorganisms. The word “antimicrobial “was replaced by “antibacterial” as suggested by the reviewer.
- Abstract: Line no. 29, “3 seasons” corrected into “three seasons”. Line 29 has been corrected as suggested by the reviewer.
- The table caption for figure 4 should be corrected. How author mentioned the sum….. of the ratio? The table caption was corrected. Also, in attention to another reviewer the table 3 and 4 were joined.
- Table 5. Why wild oregano was not compared with all three seasons? Please explain and correct Winter O. We appreciate the comment. Wild oregano was collected only in autumn, since it is the period when the highest oil yields. The main aim of this work was to evaluate the quality in terms of yield, thymol and carvacrol content as well as the antioxidant activity of the essential oil of a cultivated oregano Poliomintha longiflora Gray during the winter, summer, and autumn seasons, as well as make a comparison of cultivated oregano with wild oregano in autumn since this is the rainy season and therefore, the period when the highest yields of oregano have been reported (Lines 120 - 123). Additionally, was corrected Winter O in table 3.
- The zone of inhibition is very high, and the oil is loaded into the 6 mm disks. How was the zone of inhibition calculated? The authors suggested providing visual evidence for the antibacterial activity of the oil.
We appreciate reviewer comment. In these specific cases (B. cereus and L. monocytogenes) the halos were very large then overlapping, in all these cases we decided reported > 5.4cm. The result > 5.6 was changed because 5.4 was the largest diameter that could be measured. This was already clarified in results section (line 348 section 2.5) and corrected in the table 5.
- Overall, the comparison is useful for that condition. But those finding are not considered useful widely. If so, please explain in the introduction. The explanation is in the introduction section on lines 89 to 101. “Worldwide, Mexico is the second largest producer of oregano, mainly with the species L. graveolens Kunth, berlandieri Schauer, and recently, with the species P. longiflora Gray [8,10,23], but due to the fact that most of the commercially exploited species are of wild origin and depend on the rainy season for their harvest and are also affected by the increasing climatic changes such as long dry seasons [5,23], neither the supply or oregano nor the quality of essential oil in its composition is ensured. Moreover, the natural recovery of this resource is not guaranteed, so one of the strategies to avoid the decrease of wild oregano populations is the use of commercial crops. The production of cultivated oregano would generate changes in the composition of the essential oil with respect to wild oregano but would allow to ensure the supply and the possible obtaining of a standardized oil, so it is necessary to evaluate the yield and quality of the essential oil of cultivated oregano [24] to potentially supply the increasing demand for bioactive compounds from essential oils [10].”
- In conclusion, the author mentioned that “The main difference found was the ratio of thymol: carvacrol in wild oregano oil was 1:8.6 while in cultivated oregano, it was approximately 1:2 which was maintained in all three seasons”. How was it calculated? It is important to indicate the thymol:carvacrol synergies, in antibacterial and antioxidant activities, in this sense it is relevant to compare the ratio of these in the oil as well as the %w/w of thymol and carvacrol individually. The ratio was obtained from the amount of carvacrol divided by the amount of thymol (Ratio=carvacrol/thymol).

Reviewer 2 Report
Manuscript ID: plants-1768668
Type of manuscript: Article
Title: Comparison of chemical composition, physicochemical parameters,
antioxidant, and antimicrobial activity of the essential oil of cultivated
and wild Mexican oregano Poliomintha longiflora Gray
In the present article, the authors studied the quality in terms of yield, thymol and carvacrol content, the antioxidant activity of the essential oils of a cultivated oregano Poliomintha longiflora Gray collected during the winter, summer, and autumn seasons, and they compare the results with those of wild oregano collected in autumn, the period when the highest oil yields. The manuscript, while not scientifically original, is well written and the experiments were well planned. I recommend the acceptance of the paper after minor revisions listed below.
Abstract
Line 20, page 1: Get rid of the comma after “conditions”.
Please, provide more incisive Keywords
Introduction
Line 54, page 2: Get rid of “in its composition”.
Line 61 page 2: The sentence “and to the presence of hydroxyl groups in these phenolic compounds” is not clear.
Results and discussion
Line 96, page 3: Please explain why you have used anethole oil as a reference and include reference.
Table 5, page 5: You have indicated in the text, in the comments to the table, the major components of essential oils resulting by GC-MS analyses, but you have made no mention about the "other components" which by the way are 24.27% in winter and about 16% in the other seasons. Please provide some information about them.
Lines 182-188, page 5: The sentence is too long. Please rewrite with shorter and clearer sentences
Materials and methods
3.1. Plant Material:
Were the plants collected on the same day or over a longer period?
Why did you not collect oregano Poliomintha longiflora Gray from more areas of northern part of Mexico to make your conclusions more accurate, since the quality and yield of oregano is strongly influenced by soil and climatic conditions and varies from area to area?
Author Response
RESPONSE TO REVIEWER 2
Dear Reviewer #2.
We appreciate your valuable time and comments on the manuscript entitled “Comparison of chemical composition, physicochemical parameters, antioxidant, and antimicrobial activity of the essential oil of cultivated and wild Mexican oregano Poliomintha longiflora Gray”.
Abstract
Line 20, page 1: Get rid of the comma after “conditions”.
Please, provide more incisive Keywords
This was corrected as suggested by the reviewer. Also, more keywords were provided (wild oregano; cultivated oregano, thymol, carvacrol).
Line 54, page 2: Get rid of “in its composition”. Line 68 has been corrected as suggested by the reviewer.
Line 61 page 2: The sentence “and to the presence of hydroxyl groups in these phenolic compounds” is not clear. The lines 75-76 were restructured to be clearer as suggested by the reviewer, as shown below:
The antioxidant and antibacterial activities can be explained by the predominance of compounds such as thymol and carvacrol, which have hydroxyl groups.
Results and discussion
Line 96, page 3: Please explain why you have used anethole oil as a reference and include reference.
The anethole oil is a reference standard in United States pharmacopeia, thus we used it was a control to verify that the equipment measures the refractive index correctly; the reference values of the anethole standard RI 1.56. In addition, the RI and density reported Maree and Ian (2000) [25] for B. anisata oil (89.9% anethole) were relative density 0.984 and refractive index 1.5572, the values are like those obtained for the anethole. This information was clarified in the lines 120-123.
Table 5, page 5: You have indicated in the text, in the comments to the table, the major components of essential oils resulting by GC-MS analyses, but you have made no mention about the "other components" which by the way are 24.27% in winter and about 16% in the other seasons. Please provide some information about them.
We appreciate your valuable comments, however, as indicated in the introduction, lines 75 - 76, the antioxidant and antimicrobial activities, which are the ones studied in this research, are attributed to a group of compounds, monoterpenes, including thymol and carvacrol. Due to aforementioned, the instrumental conditions of analysis were focused on detecting these group of compounds, which are the main interest as explained above.
Lines 182-188, page 5: The sentence is too long. Please rewrite with shorter and clearer sentences. The sentence was rewritten as suggested by the reviewer. As shown below:
Lines 267-271, “In a study conducted by Sarrazin et al. [32], the composition in oregano Lippia origanoides was reported in the rainy season and dry season, carvacrol was the main component with 43.5% and 41.4% followed by thymol with 10.7% and 10.6%, ρ-cymene with 9.8% and 10,0% as well as ρ-methoxythymol with 9.6% and 10.4% respectively for a total of 78.6% of the oil composition and showing little difference between seasons.”
Materials and methods
3.1. Plant Material:
Were the plants collected on the same day or over a longer period?
In order to clarify the information, the following paragraph was added in lines 397-402:
“The aerial parts of the plants were collected in a single day in the winter (January), summer (June) and autumn (September) seasons. Additionally, the wild type was sampled on the same day as the cultivated from the autumn season, in all cases the samplings were carried out in the same municipality”.
Why did you not collect oregano Poliomintha longiflora Gray from more areas of northern part of Mexico to make your conclusions more accurate, since the quality and yield of oregano is strongly influenced by soil and climatic conditions and varies from area to area? The scope of this study was to carry out a comparative study between cultivated and wild oregano, the latter located within an area close to the cultivated one. This is explained in point 3.1, section 3.0 Materials and Methods.

Reviewer 3 Report
The results presented in the manuscript are of a practical nature and may be published. However, I have a number of comments. It is not clear what the authors mean by "physicochemical composition" and what meaning "relative densities and the refractive indices" mean. If these are really important indicators, then they should be discussed. However, in my opinion, sections 2.1 and 2.3 should be merged. Tables 1 and 2 can be combined. Tables 3 and 4 should be combined. Table 6 needs to be corrected. If the data in Table 6 are presented as the mean and the statistical error of the mean (SEM), then the SEM in column 2 is incorrect.
It is not clear why the authors did not provide full data on antimicrobial activity, but limited the inhibition zone to "5.6 cm". "Results are presented as a function of diameters of inhibition zones in cm" should be removed from Table 7 title.
Author Response
Dear Reviewer #3.
We appreciate your valuable time and comments on the manuscript entitled “Comparison of chemical composition, physicochemical parameters, antioxidant, and antimicrobial activity of the essential oil of cultivated and wild Mexican oregano Poliomintha longiflora Gray”.
-It is not clear what the authors mean by "physicochemical composition" and what meaning "relative densities and the refractive indices" mean. "physicochemical composition" was replaced by "physicochemical properties" similarity “Relative densities and refractive indices" were replaced by "Density and refractive index.” As shown in lines 126-127.
-If these are really important indicators, then they should be discussed. Ok, information regarding these indicators was included in text (lines 136 -137).
-However, in my opinion, sections 2.1 and 2.3 should be merged. Tables 1 and 2 can be combined. Tables 3 and 4 should be combined.
We agree. Sections 2.1 to 2.3 were merged. Also, Tables 1 and 2 and Tables 3 and 4 were combined as suggested by the reviewer.
-Table 6 needs to be corrected. If the data in Table 6 are presented as the mean and the statistical error of the mean (SEM), then the SEM in column 2 is incorrect.
The data in Table 6 are presented as the mean and standard deviation not as relative standard error and this was clarified as a footnote.
-It is not clear why the authors did not provide full data on antimicrobial activity, but limited the inhibition zone to "5.6 cm". "Results are presented as a function of diameters of inhibition zones in cm" should be removed from Table 7 title.
We appreciate the reviewer comment. The sentence was removed from the title, in addition, the scope in this phase of the study is to visualize in a general way the antibacterial activity of the types of cultivated and wild oregano, thus, the inhibition ring method fulfills this purpose.

Round 2
Reviewer 1 Report
The revised manuscript was improved a lot compared to the original version. However, the author did not clarify one of the previous comments satisfactorily.
Comment
The zone of inhibition is very high, and the oil is loaded into the 6 mm disks. How was the zone of inhibition calculated? The authors suggested providing visual evidence for the antibacterial activity of the oil”. Still, I have not convinced that how disc-loaded oregano oil (10 µL) exhibits a higher zone of inhibition. Hence, the authors requested to be reevaluated the antibacterial results and provide visual evidence.
Author Response
Reviewer 1
The revised manuscript was improved a lot compared to the original version. However, the author did not clarify one of the previous comments satisfactorily.
RESPONSE TO REVIEWER 1
Dear Reviewer #1.
The zone of inhibition is very high, and the oil is loaded into the 6 mm disks. How was the zone of inhibition calculated? The authors suggested providing visual evidence for the antibacterial activity of the oil”. Still, I have not convinced that how disc-loaded oregano oil (10 µL) exhibits a higher zone of inhibition. Hence, the authors requested to be reevaluated the antibacterial results and provide visual evidence.
The authors appreciate the reviewer´s comment. We apologize because we miss to attach the photograph, but Is attached now. In these specific cases (B. cereus and L. monocytogenes) the halos were very large. The diameter of inhibition zones surrounding the disks was measured with a vernier caliper (line 513). In some cases, they converge, and in others, they caused a complete bacterial inhibition (photographic evidence is attached), thus, we reported inhibition zones >5.4cm (Table 5).

Reviewer 3 Report
The manuscript can be publiched.
Author Response
We appreciate the reviewer's comment.
Round 3
Reviewer 1 Report
-